



# Root growth dynamics and allocation as a response to rapid and local changes in soil moisture

Samuele Ceolin[1,2], Stanislaus J. Schymanski[1,2], Dagmar van Dusschoten[3], Robert Koller[3], and Julian Klaus[4]

[1]Environmental Research and Innovation (ERIN), Luxembourg Institute of Science and Technology (LIST), Esch-sur-Alzette, Luxembourg
[2]Faculty of Science, Technology and Medicine, University of Luxembourg, Esch-sur-Alzette, Luxembourg
[3]Institute of Bio- and Geosciences, Plant Sciences (IBG-2), Forschungszentrum Juelich GmbH, 52425 Juelich, Germany
[4]Department of Geography, University of Bonn, Bonn, Germany

**Correspondence:** Stanislaus J. Schymanski (stanislaus.schymanski@list.lu)

**Abstract.** Roots exhibit plasticity in morphology and physiology when exposed to fluctuating nutrient and water availability. However, the dynamics of daily time-scale adjustments to changes in water availability are unclear and experimental evidence of the rates of such adjustments is needed. In this study we investigated how the root system responds within days to a sudden and localized increase in soil moisture ("Hydromatching"). Root systems of maize plants were grown in soil columns divided into four layers by vaseline barriers and continuously monitored using a magnetic resonance imaging (MRI) technology. We found that within 48 hours after application of water pulses in a given soil layer, root growth rates in that layer increased, while root growth rates in other layers decreased. Our results indicate local root growth was guided by local changes in soil moisture and potentially even by changes in soil moisture occurring in other parts of the soil profile, which would result in a coordinated response of the entire root system. Hydromatching in maize appears to be a dynamic and reversible phenomenon, for which the investment in biomass is continuously promoted in wet soil volumes and/or halted in drier soil volumes. This sheds new light onto the plasticity of root systems of maize plants and their ability to adjust to local and sudden changes in soil moisture, as would be expected due to patchy infiltration after rainfall or irrigation events.

## 1 Introduction

Plant water uptake can be a key limitation to plant growth, above-ground net primary productivity and crop yields. Droughts are expected to increase in frequency, duration and intensity in the future with potentially severe effects on vegetation (Van Loon et al., 2016; Klaus et al., 2022). Crop yields that are sensitive to water shortage are expected to decline in the future and research to increase drought tolerance and root water uptake efficiency is intensifying (Chaves and Davies, 2010; Daryanto et al., 2017; Farrant and Hilhorst, 2022). One promising mitigation strategy to cope with increased drought stress is to implement plant breeding programs that enhance water uptake through root traits selection (Eshel and Beeckman, 2013).

Roots are known to display plasticity in morphology and physiology under environmental fluctuations (Hodge, 2004; Fromm, 2019). In the presence of spatial variability in soil moisture, plants rely on compensatory mechanisms to meet their transpiration



demand. These mechanisms consist of enhanced water uptake from zones with higher moisture that compensate for the lower uptake from the drier zones (Thomas, 2020). In the short-term (hours), local adjustments in uptake rates are mostly driven by water potential gradients (Jarvis, 2011). It can be assumed that the root-soil water potential difference drives root water

uptake, while the radial root conductivity and root surface area determine the main resistance to water uptake in relatively wet soils (Schymanski et al., 2008). At the same time, root hydraulic conductivity can be regulated by aquaporin activity locally at sub-hourly time scales to promote water uptake in water-rich areas (Carvajal et al., 1996; Gorska et al., 2008; Ishikawa-Sakurai et al., 2014).

On a longer term (days), root systems can undergo morphological adjustments consisting of promotion of root growth within

wetter soil zones, increasing the surface area available for water adsorption (Pregitzer et al., 1993; Majdi and Andersson, 2005; Wang et al., 2005; Bauerle et al., 2008; Zhang et al., 2019; Tzohar et al., 2021). Among such morphological adjustments of roots, we find "Hydrotropism" and "Hydropatterning" that refer to root curvature towards zones of high water potential and to the promotion of asymmetrical lateral root formation towards wet soil patches, respectively. "Xerobranching", in contrast, refers to the inhibition of lateral root development in dry soil areas (Giehl and von Wirén, 2018; Fromm, 2019).

Studies also reported another type of morphological adjustment consisting of root proliferation within a wet soil patch and in decline of root growth in drier areas (Engels et al., 1994; Gallardo et al., 1994). As a combination of these latter responses has the capacity to adjust the entire root system to spatial variation in soil water availability, we call this process "Hydromatching" for ease of reference. Hydromatching differs from Hydrotropism, Hydropatterning and Xerobranching as it does not dictate the direction of individual root growth or lateral root emergence. Instead, it considers a larger scale and includes changes to both

the elongation rate of pre-existing roots and emergence rate of new roots (of any order) from portions of root systems experiencing a change in soil moisture. Under temporal heterogeneity in soil moisture, studies have mostly focused on seasonal time scales and have shown that plants are able to dynamically shift their root length distribution in response to seasonal fluctuations (Hayes and Seastedt, 1987; Wan et al., 2002; Peek et al., 2006; Saelim et al., 2019).

Although the morphological adjustments under spatial soil moisture changes and the root growth dynamics to seasonal oscilla-

tions are well studied, little is known about the dynamics and flexibility of root growth patterns under rapid (daily) changes of soil moisture. Such daily patterns are in fact often overlooked (Stewart and Frank, 2008) and can strongly affect plant fitness (Nguyen et al., 2017). In addition, studies have yet to prove the ability of plants to deploy morphological adjustments according to multiple abrupt, daily changes in soil moisture in different parts the soil profile. A better understanding of such dynamics could lay the groundwork to improve irrigation management and the performance of soil-vegetation-atmosphere models. In

fact, it was already shown that models with dynamic root systems performed better than models with static root systems (Schymanski et al., 2008; Wang et al., 2018).

The overarching aim of this study is to decipher the dynamics of the Hydromatching phenomenon at daily time scale. Specifically, we build on previous studies (Gallardo et al., 1994; Engels et al., 1994; Wang et al., 2005) by analyzing the Hydromatching dynamics such as timing of occurrence, depth independence and local root allocation patterns following abrupt changes in

soil moisture. We monitored these dynamics following rapid spatial (different depths) and temporal changes in soil moisture induced by consecutive water pulses applied in different soil layers. The following research questions were addressed:



– Question 1 (onset time of Hydromatching): How quickly does Hydromatching occur following a rapid and local change in soil moisture availability?

– Question 2 (vertical responsiveness): Do roots at different depths respond equally when subjected to local changes in soil moisture availability?

To address these questions we designed an experiment involving the use of maize plants grown under highly controlled conditions in a horizontal split-root setup consisting of layered soil columns. Root development was monitored at a high frequency (every two days) using a Magnetic Resonance Imaging (MRI) technology. This produces 3D images of root systems and allows to measure root distribution repeatedly and non-destructively.

## 2 Materials and methods

### 2.1 Experimental setup

Maize (*Zea mays*) was used as the species of interest, as its roots are well suited for MRI detection (Müllers et al., 2023) and pre-experiments showed that maize grows well in our experimental setup. The maize plants were grown in 45 cm long plexiglass tubes filled with soil (refer to next section for soil specifics) and divided in four layers (top three layers with 9 cm depth and the bottom layer with 14 cm depth). The layers were hydraulically isolated from each other by vaseline barriers. In the below, we refer to the top layer as Layer 1 (L1), followed by Layer 2 (L2) and so on. Each layer was equipped with a "rhizon", a small tube made of a porous material designed for water sampling in soil (10 cm length, 0.15 $\mu$m pore size; Rhizon MOM 19.21.21, Rhizosphere Research Products). Rhizons were used to inject water in the soil layers. The vertical soil moisture distribution was recorded using the Soil Water Profiler (SWaP) (van Dusschoten et al., 2020). This sensor allows the determination of volumetric soil moisture (VWC) profiles with a one-dimensional vertical resolution of 1 cm. Measurements of volumetric water content allowed calculations of water uptake in each layer.

### 2.2 Preparatory stage

Seeds with a weight between 0.36 and 0.46 g were placed in wet filter paper in the dark at 20 °C for four days prior to planting. Subsequently, two seeds were planted per column at a depth of 0.5 cm and covered with 2 cm of soil, for a total of 24 columns. Silty sand soil (LUFA Speyer 2.1, LUFA, Speyer, Germany) was used and it was packed at a bulk density of 1.48 g cm$^{-3}$ in each layer. Tap water was added to the soil to reach 15% volumetric water content. The added water contained 1.5 g l$^{-1}$ of NPK fertilizer (20% Nitrogen Total, 20% Phosphorus Pentoxide ($P_2O_5$), 20% Potassium Oxide ($K_2O$) and trace elements, Allrounder Peters Professional, ICL), resulting in 0.1 g of fertilizer per layer at the start of the experiment. The soil surface was covered with perforated parafilm to reduce evaporation while ensuring adequate aeration and limiting mold formation. Soil columns were placed in a room at 30 °C to boost seedling growth. Once one of the sprouts grew through the parafilm (approximately after three days) the seed with the shorter sprout was removed. The columns were transferred to a room at 24 °C and exposed them to a light/dark period of 10/14 hours. In order to provide the same light intensity to each plant, each soil



column was placed inside a 90 cm high white PVC column with an outer diameter of 15 cm. The top of the PVC columns was open to allow air circulation. The PVC columns were elevated by 10 cm so that air could also circulate from the bottom upwards. Each PVC column had a LED mounted on top. The LED was programmed to produce a PAR of 600 $\mu$mol s$^{-1}$ m$^{-2}$ at 30 cm above the soil surface (approximately halfway between the soil surface and the light source). This height corresponds to the height that previously tested plants reached after a period of three weeks. Each column was weighed every second day to determine the amount of transpired water. The amount of transpired water was replenished in the top layer following the weighing. Two weeks after sowing, the plants were placed in a well ventilated growth room with a temperature range of 21-23 °C, a humidity of 60% and the same lighting conditions as before.

### 2.3 Core experimental stage

Eighteen plants (out of 24) that performed the best in terms of shoot growth, transpiration and root abundance were selected to efficiently manage the workload. These plants were allowed to acclimate and grow their root systems under 15% VWC maintained in each layer for another two weeks. Whenever the amount of water to be replenished in the top layer exceeded 20 ml, watering occurred directly by pouring water from the top to speed up the process. Water uptake rates were inferred from the decline in VWC since the water replenishment on the previous day.

At the end of the acclimation period (four weeks after sowing) plants were allowed to draw down soil moisture without replenishing it to induce water stress for six days. To avoid excessive damage to the root system, VWC was not allowed to remain below 6% VWC in any layer, i.e. water was replenished back to 6% VWC whenever it declined below this value. After the six days of water stress we grouped the plants in two treatments and one control, where VWC was raised again to 15% in all soil layers and maintained at this level until the end of the experiment. Treatment 1 (T1) group received a first series of water pulses in Layer 2 (L2) followed by a second series of pulses in Layer 1 (L1). Treatment 2 (T2) group received the pulses in reverse order. A series of pulses consisted of one water injection per day in a given layer to reach 15% VWC. The experiment was divided into three phases, each lasting four days (Fig. 1): Phase 0 (drought period), Phase 1 (period of the first series of pulses) and Phase 2 (period of the second series of pulses). This means that Layer 1 and Layer 2 both received one series of pulses, either during Phase 1 or Phase 2 depending on the treatment group (T1 or T2). Phase 0 consisted of the four days prior to Phase 1, i.e. the water stress period. Phase 1 started on the fifth day after the start of Phase 0. Phase 2 started once the VWC of the layer treated during Phase 1 dropped below 7.5%, i.e. a few days after the discontinuation of the pulse. This means that Phase 2 started at different times for each plant, depending on its actual water uptake rates (refer to Fig. S2.1 in the Supplementary information document). The experiment ended once the VWC in the layer treated during Phase 2 was depleted to half its maximum VWC value. For simplicity, "a series of pulses" is referred to below as "pulse".

### 2.4 MRI and MRI image analysis

The experiment was carried out using the PlantMRI installation at IBG-2 at the Forschungszentrum Jülich (FZJ). The PlantMRI is suitable for repetitive and non-destructive 3D imaging and measurement of root traits. MRI uses the magnetic moment of atomic nuclei such as H$^1$ (protons), which are highly present in water and hence in living tissues. The technology relies on





**Figure 1.** Summary of the treatments in the two treatment groups and of the temporal organization into phases in treatment and control groups. During Phase 0 we applied no water, while during Phase 1 and Phase 2 water pulses were applied in different layers depending on the treatment considered.

magnetic and radio frequency fields and contrast parameters to differentiate between roots and the background (van Dusschoten et al., 2016). The 18 selected plants were sub-divided into two groups and their root profiles were MRI-scanned every day on one group to accommodate the MRI imaging schedule. As a result, we obtained images daily but each specific root system was imaged every two days. The software NMRooting was then applied for root trait analysis. (van Dusschoten et al., 2016).

In many images the estimated root length in Layer 1 (L1) was prone to artifacts and a global analysis of the layer resulted in erroneous estimates. When watering from the top exceeded 20 ml, water signal in soil became visible and induced artifacts that affected root trait analysis. To overcome the problem of these artifacts, we selected and measured the root length of individual roots in L1 which signal was undisturbed. To limit our selection and analysis to roots that showed some growth during the




treatments, we compared the root systems right before Phase 1 and at the end of Phase 2. Layer 2 (L2) was less prone to
130 artifacts and here total root length was measured within the whole layer.

## 2.5  Data analysis

Growth rates (mm d$^{-1}$) of each selected root in L1 and of each root system portion in L2 were calculated by subtracting the
previous root length measurement from the subsequent measurement, divided by days between measurements. Comparability
of growth rates between different plants and between layers was enabled by normalizing each growth rate time series by its
maximum value ("scaled growth rate"). We calculated then the median of the scaled growth rates in L1 and in L2 in each
treatment group and control. Since the imaging of each root system was performed every two days, each layer had two data
points in each phase of the experiment. Measurements of volumetric water content were used to calculate water uptake in each
layer as ml d$^{-1}$.

The non-parametric Mann-Whitney U test was used in our analysis to compare groups of scaled growth rates and assess
if their distributions differed significantly. This test was part of the package "stats" of the Python library "scipy" (version
1.6.0) (Virtanen et al., 2020). The test was used to evaluate the occurrence of Hydromatching as a response to our treatments
by comparing the growth rates before and after the pulse. Question 1 (onset time of Hydromatching) was then address by
comparing the growth rates before and two days after the start of the pulse in each layer. Question 2 (vertical responsiveness)
was tackled by comparing the growth rates before and after the pulse separately for L1 and L2. We also compared the growth
rates in L1 with the ones in L2 within the same treatment group and within the same phase to estimate root allocation switches
between layers from Phase 1 to Phase 2.

Hydromatching was considered to have occurred when growth rates increased in a pulsed treatment layer and/or when growth
rates decreased in a non-pulsed treatment layer.

To assess whether root growth was only affected by local soil moisture or also by changes in soil moisture in a neighbouring
soil layer, we carried out correlation analyses between these covariates. The correlations were analysed between scaled growth
rates (of each individual root in L1 and of the whole root system portion in L2 of each treated plant) and the volumetric water
content (VWC) in their own layer and between the scaled growth rates and the VWC in the other layer. We used the Pearson
correlation coefficient when both sample groups were normally distributed. We used the Spearman rank-order correlation
coefficient when at least one of the two sample groups was not normally distributed. We considered the correlation significant
if the p-value was below 0.05. Both correlation functions are part of the package "stats" of the Python library "scipy" (version
1.6.0). We ended up performing 60 correlations of scaled growth rates vs local VWC and 60 correlations of scaled growth rates
vs VWC in the other layer. Additionally, we visually investigated the evolution of volumetric water content (VWC, %), water
uptake rates (ml d$^{-1}$) and scaled growth rates in time series plots in order to shed light on potential links between root growth
dynamics and water acquisition.





# 3 Results

## 3.1 Occurrence of Hydromatching

The occurrance of Hydromatching was determined by comparing the scaled growth rates between phases for the treatment layers pulsed during Phase 1 (L2 of T1 plants and L1 of T2 plants, see Fig. 1) and for the treatment layers pulsed during Phase 2 (L1 of T1 plants and L2 of T2 plants). The scaled growth rates of treatment layers pulsed during Phase 1 significantly increased from Phase 0 (median of 0.06) to Phase 1 (median of 0.55) and significantly decreased in Phase 2 (median of 0.01, purple line in Fig. 3, Table 1, Fig. 2). The scaled growth rates of treatment layers pulsed during Phase 2 significantly increased from Phase 1 (median of 0.01) to Phase 2 (median of 0.61, brown line in Fig. 3, Table 1, Fig. 2). The scaled growth rates in both layers of the controls increased significantly from Phase 0 (median of 0) to Phase 1 (median of 0.74) and significantly decreased in Phase 2 (median of 0.45, black line in Fig. 3, Table 1). These results indicate that growth rates in a layer increased markedly after transitioning from non-pulsed to pulsed in both treatments and controls. Growth rates in treatment layers also decreased markedly after transitioning from pulsed to non-pulsed. In the controls, growth rates decreased in Phase 2 even though volumetric water content (VWC) was kept at the same level as in Phase 1.

During Phase 1 the scaled growth rates in the controls and in the pulsed treatment layers did not differ significantly (Table 2), but were both significantly higher than the scaled growth rates in the non-pulsed treatment layers (namely the layers pulsed in Phase 2, Table 2). The same was found for Phase 2. During Phase 1 the median of the scaled growth rates was 0.74 for the controls, 0.55 for the pulsed treatment layers and 0.01 for the non-pulsed treatment layers (Fig. 3 and Table 1). During Phase 2 the median of the scaled growth rates was 0.45 for the controls, 0.61 for the pulsed treatment layers and 0.01 for the non-pulsed treatment layers (Fig. 3 and Table 1). The fact that the scaled root growth rates in the control layers were similar to those in pulsed treatment layers and significantly greater than those in non-pulsed treatment layers indicates that the treatments were responsible for the changes in growth rates.

Note that the median lines in Fig. 3 are more representative of the behavior of roots in L1. This was due to the much higher availability of data from L1 than from L2 (5 data points per plant per day from L1 versus only 1 data point per plant per day from L2). This explains the strong resemblance of the median lines in Fig. 3 with the median lines in Fig. 4a and c.

## 3.2 Onset time of Hydromatching after pulse

We compared the scaled growth rates in both pulsed treatment layers and control layers on the first day of pulse and 48 hours after to determine the onset time of Hydromatching. The scaled growth rates increased significantly during this time interval (median increased from 0.05 to 0.75, p-value<0.01 not shown in tables). Refer to Fig. S2.4, S2.5 and S2.6 in the Supplementary information document for the growth rates over time in each individual plant. This, along with the evidence described at the end of subsection 3.1, implies that Hydromatching occurred within two days from a pulse. Note that in this analysis scaled growth rates of control layers and of treatment layers pulsed during Phase 1 and Phase 2 were grouped together (refer to Fig. S2.4, S2.5 and S2.6 in the Supplementary information to see what data were considered in this analysis).





Note that we experienced recurring technical difficulties with the automatic robot arm carrying the plants into the MRI. For example, the robot arm would sometimes get jammed after only carrying one plant into the MRI machine and would stop working for the rest of the imaging sessions, which were scheduled at night. In such case, only one plant was imaged on that

day and the rest of the plants had to be imaged the following morning. This explains why in Fig. 3 and Fig. 4 there are days with only one measurement of growth rate available. These technicalities also did not allow us to maintain a constant 48 h interval between imaging events for each root system, which, at times, occurred 24 and 72 hours apart. In any case, growth rates ($d^{-1}$) were calculated by dividing the change in root length between two measurements by the time interval between measurements.

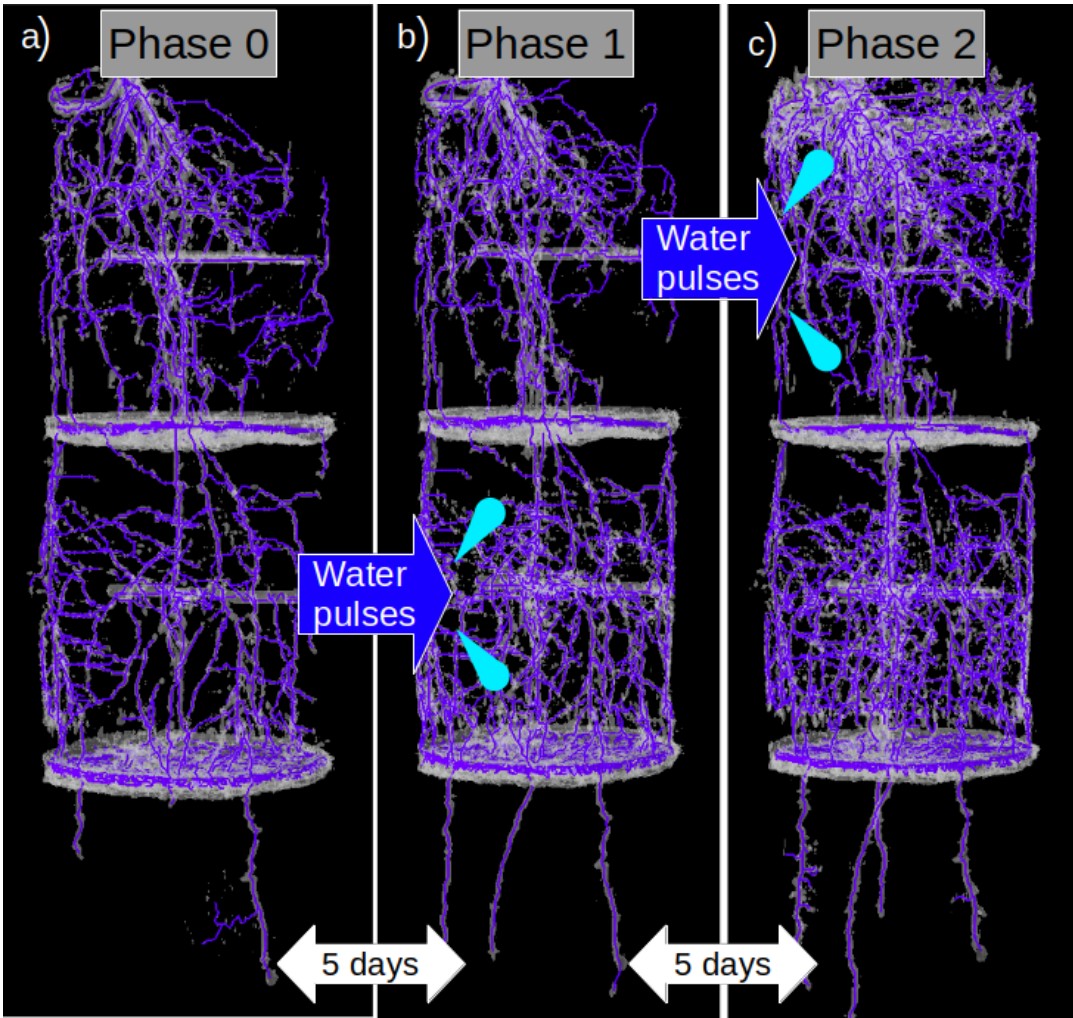

**Figure 2.** Selected MRI images of the same root system at the end of Phase 0 (a), in the middle of Phase 1 (b) and in the middle of Phase 2 (c). A water pulse was a applied in Layer 2 during Phase 1 and another pulse was later applied in Layer 1 during Phase 2. An increase in the root abundance in the pulsed layer is observable during both Phase 1 and Phase 2.





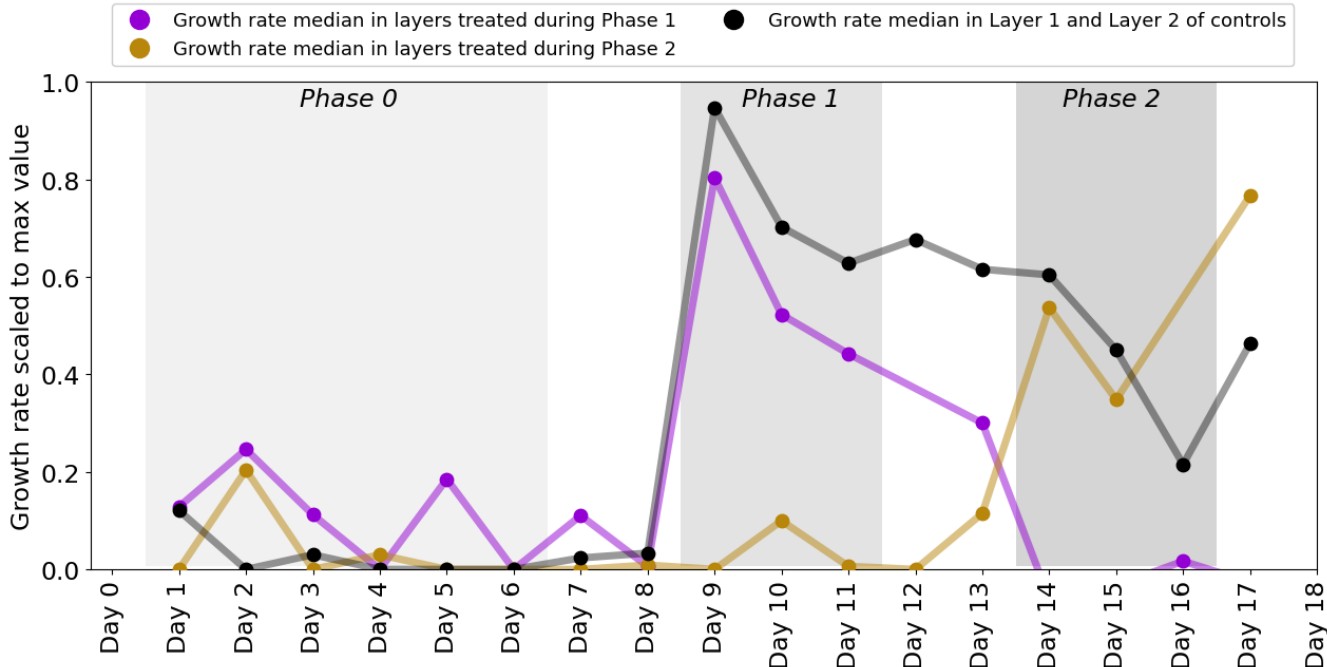

**Figure 3.** Median of the scaled growth rates in the treatment layers pulsed during Phase 1 (Layer 1 in Treatment 2 plants and Layer 2 in Treatment 1 plants), Phase 2 (Layer 1 in Treatment 1 plants and Layer 2 in Treatment 2 plants) and in the controls (where both layers were kept at 15% VWC during Phase 1 and 2). Each median point was calculated from sets of daily data which size ranged from 5 to 29 data points. Each median point contains mixed information of both treatments and layers. The medians of the control group contain mixed information of both layers. Only single data points of scaled growth rate were available on Day 12 for the layers treated during Phase 1 (brown line) and on Day 16 for the layers treated during Phase 2 (purple line). These values were excluded from the plot. The areas of different gray shades indicate the three phases containing the data points used in the analysis. Negative scaled growth rates are not shown. Differences in number of data points per measurement day are due to technical limitation.





**Table 1.** Medians of the scaled root growth rates (with 25th and 75th percentiles) and p-values of the comparisons between the scaled growth rates during different phases for treatment layers pulsed during Phase 1, treatment layers pulsed during Phase 2 and control layers. For the controls, medians and scaled growth rates used in the comparisons refer to both Layer 1 and 2. P-values indicate the probability that an observed difference between the groups considered is coincidentally sampled from the same distribution.

|  | Treatment layers pulsed in Phase 1 | Treatment layers pulsed in Phase 2 | Layers of controls |
|---|---|---|---|
| Median of scaled growth rates in Phase 0 (25th - 75th percentile) | 0.06 (0 - 0.23) | 0 (0 - 0.22) | 0 (0 - 0.16) |
| Median of scaled growth rates in Phase 1 (25th - 75th percentile) | 0.55 (0.27 - 1) | 0.01 (0 - 0.21) | 0.74 (0.42 - 1) |
| Median of scaled growth rates in Phase 2 (25th - 75th percentile) | 0.01 (−0.11 - 0.19) | 0.61 (0.19 - 1) | 0.45 (0.25 - 0.75) |
| Scaled growth rates in Phase 0 vs Phase 1 (p-value) | <0.01 | 0.38 | <0.01 |
| Scaled growth rates in Phase 1 vs Phase 2 (p-value) | <0.01 | <0.01 | <0.01 |

**Table 2.** P-values of the comparison of the scaled growth rates of the control layers with the scaled growth rates of the non-pulsed treatment layers, from the comparison between control layers and pulsed treatment layers and from the comparison between pulsed and non-pulsed treatment layers. The comparison was done between scaled growth rates within the same phase. Refer to Table 1 for the medians of the scaled growth rates during each phase.

|  | Scaled growth rates in controls vs non-pulsed treatment layers | Scaled growth rates in controls vs pulsed treatment layers | Scaled growth rates in non-pulsed treatment layers vs pulsed treatment layers |
|---|---|---|---|
| Phase 1 | <0.01 | 0.13 | <0.01 |
| Phase 2 | <0.01 | 0.19 | <0.01 |

## 3.3 Vertical responsiveness of Hydromatching

Scaled growth rates during the pre-pulse and post-pulse were compared separately for L1 and L2 to determine whether roots at different depths would respond equally. The scaled growth rates increased significantly in L1 after the start of the treatment, both in Phase 2 of T1 plants and Phase 1 of T2 plants, with median growth rates from 0 to 0.64 and from 0.01 to 0.55, respectively (purple area in Fig. 4a and c and Table 3). However, the response of growth rates in L2 to water pulses was not statistically significant, neither in Phase 1 of T1 plants, nor Phase 2 of T2 plants (brown area in Fig. 4b and d, and Table 3),



although a small second peak is visible during Phase 2 in L2 of T2 plants (brown area in Fig. 4d). In addition to the significant increase in root growth in L1 in response to the pulse, a significant decrease in growth rate was found in non-pulsed treatment layers (both L1 and L2), when the pulse was applied in the other layer. Some growth rates reached negative values, potentially indicating root disappearance. Scaled growth rates in L1 of T2 plants significantly decreased from a median of 0.55 in Phase 1 to 0.04 in Phase 2 (purple area in Fig. 4c, Table 3). Scaled growth rates in L2 of T1 plants also significantly decreased from

a median of 0.56 in Phase 1 to -0.16 in Phase 2 (brown area in Fig. 4b, Table 3). Scaled growth rates in L2 of T2 plants decreased from a median of 0.60 in Phase 0 to 0.20 in Phase 1 (solid brown line in Fig. 4d). Although L1 clearly responded when receiving water pulses while L2 did not, we observed in both layers a significant decline in growth rate (unless already at 0) in response to a water pulse in the other layer.



**Figure 4.** Medians of scaled growth rates, volumetric water content (VWC, %) and water uptake rates (ml $24h^{-1}$) throughout the phases. (a) L1 of T1 plants. (b) L2 of T1 plants. Each median was calculated from a range of 5-20 data points per day for L1 and 2-5 data points per day for L2. Note that during the pulse VWC was increased to 15% and that the values of VWC in the plots are lower because the measurements were taken right before the watering. Only single data points of scaled growth rate were available on Day 10 and Day 12 in L2 of T1 plants (b). These values were excluded from the plot.





**Figure 4.** Medians of scaled growth rates, volumetric water content (VWC, %) and water uptake rates (ml 24h$^{-1}$) throughout the phases. (c) L1 of T2 plants. (d) L2 of T2 plants. Only single data points of scaled growth rate were available on Day 1, Day 10 and Day 16 in L2 of T2 plants (d). These values were excluded from the plot.




**Table 3.** Medians of the scaled growth rates (with 25th and 75th percentiles) and p-values of the comparisons between the scaled growth rates during different phases for the treatment layers pulsed during Phase 1 considered separately (L1 in T2 plants and L2 in T1 plants) and for the treatment layers pulsed during Phase 2 considered separately (L1 in T1 plants and L2 in T2 plants).

| | Treatment layers pulsed during Phase 1 | | Treatment layers pulsed during Phase 2 | |
|---|---|---|---|---|
| | L1 in T2 plants | L2 in T1 plants | L1 in T1 plants | L2 in T2 plants |
| Median of scaled growth rates in Phase 0 (25th - 75th percentile) | 0.01 (0 - 0.16) | 0.60 (0.31 - 0.73) | 0 (0 - 0.06) | 0.60 (0.37 - 0.75) |
| Median of scaled growth rates in Phase 1 (25th - 75th percentile) | 0.55 (0.28 - 1) | 0.56 (0.28 - 0.75) | 0 (0 - 0.12) | 0.20 (0.06 - 0.38) |
| Median of scaled growth rates in Phase 2 (25th - 75th percentile) | 0.04 ($-0.07$ - 0.26) | -0.16 ($-0.40$ - $-0.04$) | 0.64 (0.17 - 1) | 0.32 (0.20 - 0.83) |
| Scaled growth rates in Phase 0 vs Phase 1 (p-value) | <0.01 | 0.50 | 0.20 | 0.07 |
| Scaled growth rates in Phase 1 vs Phase 2 (p-value) | <0.01 | <0.01 | <0.01 | 0.14 |

## 3.4 Switches in local root growth allocation from Phase 1 to Phase 2

Scaled growth rates in L1 were compared with the ones in L2 within the same treatment group and within the same phase. This was done to estimate to what extent root growth allocation switched between layers when moving from Phase 1 to Phase 2. During Phase 1, scaled root growth rates in the pulsed treatment layers (L2 in Treatment 1 and L1 in Treatment 2) were significantly higher than in the non-pulsed treatment layers (Fig. 5 and Table 4). In Phase 2, when the water pulses were reversed, so were the growth rates, again resulting in higher growth rates in the pulsed treatment layers. This was achieved by

both decreased growth rates in non-pulsed treatment layers (observed in both L1 and L2), and increased growth rates in pulsed treatment layers (observed only in L1). This means that roots clearly switched growth allocation patterns between layers when moving from Phase 1 to Phase 2.







**Figure 5.** Growth rates in L1 and L2 during Phase 1 and Phase 2 of T1 and T2 plants. Panels on the left refer to Phase 1 and on the right to Phase 2. (a)-(b): T1; (c)-(d): T2. Each violin contains 10 data points for L2 and 50 data points for L1. The thick black line inside each violin indicates the range between the $25^{th}$ and $75^{th}$ quartiles, while the white circles mark the median values.





**Table 4.** P-values of the comparisons between the scaled growth rates in L1 and L2 during the same phase for each treatment group (T1 and T2). Refer to Table 3 for the medians of the scaled growth rates in L1 and L2 of T1 and T2 during each phase.

|  | Treatment 1 | Treatment 2 |
|---|---|---|
| Scaled growth rates in L1 vs L2 during Phase 1 | <0.01 | 0.01 |
| Scaled growth rates in L1 vs L2 during Phase 2 | <0.01 | <0.01 |

## 3.5 Links between VWC, growth rates and water uptake rates

Out of the 60 correlation analyses between scaled root growth rates and volumetric water content (VWC), 17 of them revealed
significant positive correlations between root growth and local VWC while 7 of them revealed significant negative correlations between root growth and VWC in the other layer. Out of the 50 correlation analyses between scaled root growth rates of individual roots in L1 and VWC, 5 of them revealed both a positive correlation with VWC in L1 and a negative correlation with VWC in L2.

The dynamics of responses in root growth to variations in soil moisture can be seen clearly when visually inspecting the time
series in Fig. 4. For example, it is interesting to observe how the scaled growth rates and water uptake rates in L2 of T1 plants (Fig. 4b) declined during the transition to Phase 2, when the soil moisture reached lower levels (on Day 12 and 13) and the treatment in L1 still had not been applied for any of the plants yet. In L2 of T2 plants (Fig. 4d), the scaled growth rates started decreasing during Phase 1 (when the treatment was applied in L1) and the VWC remained almost unvaried until the end of Phase 1. Water uptake rates decreased during Phase 1 and were even close to 0 ml day$^{-1}$ on Day 9. L1 of T2 plants (Fig. 4c)
showed a similar behavior as the scaled growth rates and water uptake rates dropped at the onset of Phase 2, when the treatment was applied in L2. Water uptake was lower during Phase 2 (on Day 14 and Day 15) than at the end of Phase 0 (on Day 5 and 6) even though the soil moisture was higher during Phase 2.

## 4 Discussion

### 4.1 Hydromatching observable within two days of a water pulse

Our results provide evidence that maize roots responded to a water pulse by locally increasing their growth rates within the wetted soil layer (Fig. 3). Engels et al. (1994) measured root growth twice in maize and once in rapeseed over a period of six days after the re-wetting of a top soil layer. They attributed the observed root growth promotion in the top layer to mobilization of nutrients (only present in the top layer) and not to increased soil moisture itself. In our study Hydromatching occured within 48 h after applying a water pulse in a soil layer likely without the influence of nutrients. In fact, every soil layer was supplied
with the same amount of nutrients at the beginning of the experiment, including deeper layers that always stayed relatively moister than the top layer (and where nutrients would therefore be expected to be more mobile). It is possible that changes occurred even faster than within our observation interval of 48 hours. Although enhanced local fine root production within





water patches (Pregitzer et al., 1993) and preferential root growth in watered layers over drying layers (Gallardo et al., 1994; Pardales and Yamauchi, 2003) have been documented before, here we see for the first time just how quickly roots can respond to variations in soil moisture.

## 4.2 Growth rate is continuously promoted in wetter layers and inhibited in drier layers

The increase in root growth in response to the reception of water pulses was clear in L1 but it was less evident in L2. In L2 of T1 plants growth rates remained stable while transitioning from Phase 0 to Phase 1 (Fig. 4b). This is probably because roots were still establishing in L2 and had not drawn down the soil moisture as much as in L1 by the time the water pulse was applied. In L1, the soil moisture depletion at the end of Phase 0 was so severe that root growth had stopped prior to the first pulse (Fig. 4a and c). In addition to an increase in root growth in response to pulses in L1, we observed a consistent decline in root growth in both L1 and L2 when the pulse was stopped there and was applied in the other layer (Fig. 4b and c). This suggest that maize roots at different depths respond similarly to rapid changes in soil moisture. A similar behavior has been previously documented in maize and rapeseed. For these plants, root growth increased in the top layer and decreased in the bottom layer after a water pulse at the top (Engels et al., 1994). However, our results suggest that also the opposite occurs when switching the pulse order in the layers, and that root growth decline might even reach negative values indicating root disappearance. Similar to a previous study on Kiwifruit vines (Green and Clothier, 1995), our measured water uptake rates in a layer were rapidly influenced (within 24 hours) by a change in soil moisture in both that same layer and in the neighboring layer (Fig. 4). In fact Green and Clothier (1995) also documented daily-scale shifts in uptake patterns from drier to wetter parts of the soil after re-irrigation. They suspected that this was due to a rapid flush of new root growth and reactivation of existing roots. In our case, water uptake rose prior to the increase in root growth following the water pulse (Fig. 4). We hypothesize this was caused by the sudden change in soil-root water potential gradient, which is known to drive water uptake in the short term (hours) (Jarvis, 2011). The change in gradient allowed the existing roots to markedly absorb the newly available water. Moreover, root water potential in the entire root system might become less negative when a part of the root system has access to a water source at higher water potential, according to modeling results (Amenu and Kumar, 2007; Schymanski, 2008). Hence, water uptake could decline in certain locations even though local soil moisture and canopy water demand did not change. Such situation could have occurred in our experiment as water uptake increased in the newly pulsed layer while almost coming to a halt in the other layer (Fig. 4).

In our study portions of maize root systems responded to multiple changes in soil moisture over daily (and potentially even shorter) timescales, by increasing the growth rates in wetted soil layers and/or by decreasing the growth rates in drier layers, inverting the local trends of growth rates between phases (Fig. 4 and Fig. 5). This demonstrates an exceptional level of dynamic morphological adaptation to rapidly varying moisture availability in young maize plants. The decline in root growth in response to increased soil moisture in another soil layer is one of the most intriguing findings of this study. Carbon allocation seemed to be continuously orchestrated and re-directed to match soil moisture availability, favoring the resourceful soil areas and neglecting the less beneficial ones. This behavior likely allows maize roots to chase dynamic soil moisture sources and take up enough water to meet the transpiration demand while being "cost-effective" in terms of root carbon expenditure. We suspect





that carbon allocation for root growth might be determined by both local soil moisture availability and moisture availability elsewhere. In fact, the observed root growth decline in non-pulsed layers was potentially caused by locally reduced VWC and/or even triggered by an increase in the VWC in the pulsed layer. The identification of both positive correlations between local growth rates and local VWC as well as negative correlations between local root growth and VWC in another soil layer suggests that Hydromatching may not only be a result of root growth responses to local conditions, but a consequence of a whole root system response. This coordinated response at the entire root system level could be part of a mechanism aimed at efficiently partitioning and targeting carbon allocation wherever and whenever water is more easily accessible. However, to conclusively test the hypothesis that roots respond to soil moisture variations elsewhere in the root system, soil moisture in one soil layer would need to be held constant while soil moisture in another soil layer is varied. This was not the case here, as we allowed the root system to deplete soil moisture in the non-pulsed layers.

Our findings open new avenues for research on irrigation management. The method of patchy resource supply is already known to improve efficiency in resource uptake compared to uniform resource availability, although it was mostly studied for nutrients (Fransen et al., 1999, Hutchings et al., 2003, Wang et al., 2005). When considering alternating patchy supply (changing the location of a resource patch on a daily-basis), water uptake was mostly driven by physiological changes rather than morphological changes (Fransen et al., 1999). In fact, it was hypothesized that root proliferation within resource patches may be too slow to keep up with daily changes in patch location (Van Vuuren et al., 1996, Wang et al., 2005). However, our findings indicate that maize root systems can adjust their growth rates locally within 48 hours (potentially even faster) to match rapid soil moisture changes through Hydromatching. This high degree of morphological plasticity should be further studied to assess its influence on above-ground productivity, canopy water supply and overall plant fitness. Potentially, Hydromatching could be then targeted as a breeding trait and leveraged as a tool to enhance water use efficiency under patchy water supply, caused either by differential infiltration of rainfall or drip irrigation. Vegetation models could also benefit from our findings. They could incorporate the process of Hydromatching to improve predictions of plant water use and carbon uptake, as well as soil moisture dynamics in pulse-driven ecosystems (e.g. Schwinning and Ehleringer, 2001).

## 4.3   Limitations of the study

Non-destructive root imaging was crucial for the insights obtained in this study. Due to the number of treatments and replica and the time needed for measuring one root system, each plant could only be measured every 48 hours. Since we observed root growth responses already within the first 48 hours of a treatment, it is likely that the responses occurred faster than that. To get closer to the actual response time, a similar study should be conducted with fewer individuals and more frequent imaging.

The absence (or weakness) of a response in pulsed L2 could be related to the fact that we did not leave enough time for the root systems to establish before applying the water pulses. Overall, roots reached and explored L2 later than L1. Roots in L1 had already established, substantially consumed water and halted their growth by the time Phase 0 started. Meanwhile, four plants out of ten reached L2 only at the beginning of Phase 0, which then promoted root growth due to the encounter of a wet layer (while L1 was water-depleted). This led to a smaller difference in soil moisture between the pre-pulse and post-pulse in L2 (Fig. 4b and d). VWC reached 5% in L1 of T1 plants and 4% in L1 of T2 plants before the pulse (Fig. 4a and c). In



contrast, VWC in L2 of T1 plants was still at 8% before the pulse and growth rate was promoted there already during Phase 0 because wetter than L1 (Fig. 4b). In L2 of T2 plants VWC was slightly above 6% before the treatment and here roots responded weakly to the pulse (small peak in growth rate visible during Phase 2, Fig. 4d). Overall, the initial higher levels of VWC in L2 compared to L1 may have affected the intensity of the response to the pulse in L2.

Note that a decrease in growth rates both in non-pulsed L1 and L2 suggests that roots in L2 do behave similarly to roots in L1, confirming that they likely would have increased their growth rates if pulsed following substantial water depletion. However, further testing on well-established root systems after sufficient dehydration in each layer is needed to better support this interpretation.

In connection with this, another potential limitation of the study could have been the utilization of young plants, and a replica-
325 tion of our study on mature plants would be needed to further corroborate our results. This process might not be straightforward, as more mature root systems might display different degrees of plasticity and responsiveness to soil moisture availability and changes.

Methodological challenges forced us to use different methods for root growth detection in Layer 1 (individual roots) and Layer 2 (portion of the root system). The selection of individual roots that exhibited growth throughout the entire experiment might
have introduced a positive bias towards responsive roots. Interestingly, despite the methodological differences, we found similar responses to water pulses in both layers, indicating that our findings are likely not due to a methodological artefact. On the other hand, the use of these different methods precluded us from comparing absolute root growth rates between soil layers. Future studies should consider the same method of root sample selection (individual selection or whole layer selection) for a same-scale comparison of growth rates between layers. For example, a whole layer analysis in L1 should be feasible if watering
from the top (causing artifacts) is avoided.

Additionally, targeted experiments should delve deeper into the possibility of local root growth responding to soil moisture variations in other parts of the root system. This could be done by keeping soil moisture constant in one layer while applying a pulse in a different layer.

## 5 Conclusions

In this study we observed Hydromatching in maize roots as a fast-occurring phenomenon (within 48 hours), providing robust evidence in response to Question 1 regarding the onset time of Hydromatching. This phenomenon likely enables plants to explore dynamic soil moisture sources while economizing on root carbon investments. Hydromatching occurred at different depths of maize root systems, providing evidence in response to Question 2 regarding the extent of vertical responsiveness. Root systems also showed the ability to locally switch their growth allocation and water uptake rates to match rapid changes in
soil moisture along the profile. Overall, root growth rates in maize plants were dynamically orchestrated according to temporal and spatial changes in moisture availability. Local growth was influenced by local changes in soil moisture and possibly even by changes in moisture occurring in other parts of the soil profile, which would suggest a whole-root system coordination. Targeted experiments will be needed to conclusively prove such a remote response to local changes elsewhere.



These findings are important indications for how root systems interact with their surroundings and reveal a new level of plasticity and dynamics in root systems. Future studies should use more mature root systems, established in all the soil layers considered and depleting soil moisture significantly by the time the treatments are applied. It would also be interesting to compare root dynamics of different plant species. Our findings can pave the way towards new strains of research on irrigation management. These could assess the influence of Hydromatching on canopy water supply and potentially explore its use as a tool to improve water use efficiency under patchy water supply in agricultural settings. Future studies could also incorporate our findings into vegetation-soil-atmosphere models to potentially represent a more realistic and dynamic role of root systems in affecting plant carbon and soil water fluxes.

*Code and data availability.* Data and code is available at https://zenodo.org/records/13225739.

*Author contributions.* Concept and experimental design: SC, SS, JK, DD, RK. Experimental preparation and measurements: SC, SS, DD. Data analysis: SC. Data interpretation: SC, SS, JK, DD, RK. Paper preparation: SC. All authors contributed to the discussion and revision of the paper.

*Competing interests.* The authors declare no conflict of interest.

*Acknowledgements.* This study was supported by the Luxembourg National Research Fund (FNR) - AFR programme (AFR PhD/19/SR/13577787) and by the European Plant Phenotyping Network (EPPN), project ID: 488. We would like to acknowledge Oliver O'Nagy, Frank Minette and Adriano Gama for their crucial technical input in the construction of the experimental setup. We would also like to acknowledge Dr. Daniel Pflugfelder for his support in using the MRI technology and the NMRooting software. This manuscript was written with the help of OpenAI for improved phrasing and clarity.



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
