# Peer review of "Root growth dynamics and allocation as a response to rapid and local changes in soil moisture"

_EGUsphere, 2024_

## Author Response (AR1)

**Root growth dynamics and allocation as a response to rapid and local changes in soil moisture**

Dear editor,

Thank you very much for guiding the review process and for accepting our manuscript subject to a minor revision. We have used this opportunity to substantially improve the manuscript based on the reviewer comments, and we also added a few improvements that we identified ourselves in the process, such as adding point markers to the line graphs and improving the figure captions. These additional changes were very minor, so we did not highlight them in the marked-up track-changes file. You will find all the requested changes in that file, accompanied by comments explaining why each change was made, and our responses to all the reviewer comments collected below.

Best regards,

Stan Schymanski, on behalf of Samuele Ceolin et al.

**Point by point reply to reviewers' comments**

**General comments from Reviewer 1:**

 *"Figure 3 seems to capture your results.  I'm not sure figures 4 and 5 are necessary. Figures 4 and 5 are also a little busy and difficult to interpret – the legend could be clearer.  I would recommend moving to appendices or extracting the critical data to show"*

- We moved Fig. 5 to the SI section. This figure only provides a different perspective of what is already shown in Fig. 4, so its presence is not crucial to support the points discussed later. However, we still believe Fig. 4 is key for answering Question 2 (vertical responsiveness) and to provide evidence needed to back up some important points discussed in section 4.2. The figure allows in fact to understand what happens in L1 and L2 separately in each treatment. Additionally, the figure illustrates how root growth rates varies with soil moisture. Such data visualization is needed to support our claim that water uptake increased faster than root growth,

and that soil moisture in a layer influences root growth in a different layer. Therefore, we kept Fig. 4 in the main text, but in a modified version to make it easier to interpret and to better highlight its main messages and complementarity to Fig. 3. **See these changes on page 13**

*"I think the results could be presented more succinctly to provide a clearer story for the reader (e.g., .like Fig. 3). For example, it may be clearer to combine T1 and T2 into one 'wetted' treatment"*

- We worked on improving the clarity of the results presented, for instance by clarifying the importance of differentiating between the growth rates in L1 and L2 in T1 and T2. Combining T1 and T2 works for Fig. 3 (as the goal of that figure was to show the responsiveness to the water pulse in general), but in order to answer Question 2 (vertical responsiveness) combining T1 and T2 into one "wetted" treatment would not help, as it would not allow to compare the growth rates in different layers. In order to see whether roots at different depths responded similarly when subjected to local changes in soil moisture, we need to look at each layer and at the effects of each treatment separately, because T1 and T2 differed by the order of the pulses application in the two layers (see Fig. 1). The separation also allows us to distinguish between responses to an early or a later pulse in each layer. Reporting the growth rates separately for each layer and for each treatment also allows to reveal other important insights and make some deductions, for example the fact that the lower responsiveness to the water pulse in L2 might have been given by a less intense change in soil moisture between before and after the pulse. For this reason, we kept a simplified version of Fig. 4 in the text, as described above, and explained more clearly in the text why it is important to consider T1 and T2 separately and how separate evaluations of growth rate evolution in L1 and L2 answer Question 2. **See these changes in lines 147-152 and 211-213**

**Specific comments from Reviewer 1:**

*"Figure legends are hard to interpret: change '(b) L2 of T1 plants' to a more understandable sentence / title.  I'm not sure about requirements in this journal, but I prefer some interpretation in the figures."*

- We tried to improve the clarity of the captions of Fig. 3 and 4, however the above-listed changes we applied to the figures automatically already improved the comprehensibility of the captions.

*"Figure 5 legend is mislabeled as Figure 4."*

- Now Fig. 5 belongs to the SI and the label is correct.

*"L258: 'suggests'"*

- Changed in **line 269**

*"L342: remove 'on'"*

- Changed in **line 354**

*"L350: responses in denser soils and in response to other nutrient patches would also be interesting."*

- We added a sentence addressing this point **in lines 364-365**

We also included the suggested literature. **See these changes in lines 21-24**

**Major concerns from Reviewer 2:**

*"My main concern is whether this manuscript falls within the scope of the journal (Biogeosciences). I assumed that Biogeosciences should provide meaningful information about the interactions between biological and geographical inspects. This paper mainly described biological research without strong implications for geographical things. I am probably wrong, so I would like to leave this concern to the editor and I am looking forward to reading the authors' explanation about it."*

- We chose Biogeosciences because the journal guidelines state that Biogeosciences covers "all aspects of the interactions between the biological, chemical, and physical processes in terrestrial or extraterrestrial life with the geosphere, hydrosphere, and atmosphere". They also state that the journal covers the field of plant-soil interactions. We believe that our study strongly belongs to the field of plant-soil interactions, as it involves interactions between biological processes (root development) and the geosphere and hydrosphere (local soil moisture availability dynamics) at a smaller scale (individual plant level). On top of that, root foraging abilities are known to be deeply involved in

biogeochemical processes, biomineralization and microbial weathering (processes of interest mentioned in the journal guidelines). For these reasons, we believe this manuscript falls well within the scope of Biogeosciences.

*"My second concern is about the statistical description. The authors conveyed the idea mainly by Fig. 3 with the median values and percentiles in table 1. However, given that the number of measurements at each point is not large (n=5-29), how about using mean ± std to represent the general behavior? Do you think the mean value is more representative? Can you please also include the percentile or standard deviation in Fig. 3?"*

- We used medians instead of means as our data showed skewness and a moderate presence of outliers. In this case, the median is more reliable to understand the central tendency of the data. Since adding the standard errors to median values requires bootstrapping and it is not as straightforward as for mean values, we calculated the 25$^{th}$ and 75$^{th}$ percentiles of the medians and put them in the tables to provide an idea of the median spread. We updated Fig. 3 and add the percentiles in the plot. **See these changes on page 10**

**Minor concerns from Reviewer 2:**

*"Lines 31-34: The main idea is about "Hydromatching". So do you think it is necessary to talk a lot about hydrotropism, hyrdopatterning and Xerobranching?"*

- Before introducing the term Hydromatching, we decided to describe previously documented processes of root morphological adjustments to soil moisture heterogeneity (Hydropatterning, Xerobranching), in order to provide context and differentiation. It is important to underline that, while the latter processes apply to the individual root scale, Hydromatching involves whole portions of root systems.

*"Line 75: I would explain to use the acronym more carefully. How about changing VMC to VSM? it would be difficult to link VWC to volumetric soil moisture. If you want to define for volume water content, please replace volumetric soil moisture with that."*

- We changed the instance "volumetric soil moisture" in line 75 into "volumetric water content". **See this change in line 77**

*"Lines 90-91: Can you please provide a reference about the previously tested plants?"*

- We do not really have a reference for the previously tested plants. Those were tests we performed in our lab in order to optimize the timing of the official experiment. These only consisted in growing a group of plants to assess the height they would reach in a period of approximately 3 weeks, in order to organize the timing and produce plants that were grown enough (but not too much) by the time we started our experiment in FZJ. This was clarified in the final manuscript. **See these changes in lines 93-94**

*"Line 124: As for the root length measurement, is it an output of software NMRooting? If not, can you please specify how to do that? It is very valuable for the future work."*

- NMRooting is able to reconstruct the root system in 3D according to the MRI signal, from which the Software can segment and isolate root tissues from the soil background. The reconstructed root system can then be converted into quantitative metrics such as root length: this operation is indeed a direct output of the Software. We made this clearer in the manuscript. **See these changes in lines 126-128**

*"Fig.4: Each panel has 3 y-axis lines, making it difficult to interpret. I would suggest to change the color of each y-axis line to be paired with the corresponding plots. You also can change the symbols to line with circles/dots/triangles/whatever. I do not think you need to provide all dates by x-axis. You probably can use a 2-day or 5-day interval to make the figure clearer. In addition, can you please merge 4 panels into an individual figure? It would help the readers to follow the caption."*

- After R1's suggestions, we decided to change Fig. 4 to make it easier to interpret and to better highlight its main messages and complementarity to Fig. 3 (see our response to R1's review). We decided to have two panels in Fig. 4, one for T1 and one for T2. In each panel there are three subplots, one showing the growth rates in L1 and L2, one showing the VWC in L1 and L2 and one showing the water uptake in L1 and L2. This setup allows to display the data more clearly and solve the problem of the multiple y-axes that they pointed out. This display also addresses their suggestion to use different symbols for each line: given that there are only two lines per subplot in the new Fig. 4, a simple colour distinction between them suffice to maintain clarity. The two panels were

merged into one individual figure, as they recommended. **See these changes on page 13**

*"Fig. 5: The median values in each panel (white points) are too small to follow. Can you please make it larger or mark it as a line?"*

- We marked the median values as white lines and now they are more visible. Note that Fig. 5 was moved to the SI.

*"Line 258: It should be 'suggests'."*

- Changed in **line 269**

*"Line 303: Vegetation growing/development models?"*

- We believe "dynamic vegetation models" best conveys the message. **See this change in line 314**

*"Line 305: Do you think it is possible to merge this section with the "Materials and methods" section?"*

- We understand your suggestion of moving this section into the Materials and Methods (since many of the limitations indeed arose from the methods), but we still believe this section should be placed after the discussion, as there we provide some directions on what to change in future studies to avoid the problems that we faced. Some of these problems were, for example, the lack of responsiveness in L2 and the presence of artifacts in L1 (which led us to use of two different root growth detection methods in L1 and L2). We believe that listing these limitations and potential improvements for future studies in the Materials and Methods would be premature and introduce some confusion. Discussing the limitations and potential improvements after the results have been discussed and interpreted is in our opinion more helpful, as it allows for a clearer understanding of the rationale for proposed changes to the methodology in future research.